# Laboratory Evolution of Antimicrobial Resistance in Bacteria to Develop Rational Treatment Strategies

**DOI:** 10.3390/antibiotics13010094

**Published:** 2024-01-18

**Authors:** Tomoya Maeda, Chikara Furusawa

**Affiliations:** 1Laboratory of Microbial Physiology, Research Faculty of Agriculture, Hokkaido University, Kita 9, Nishi 9, Kita-ku, Sapporo 060-8589, Japan; 2Center for Biosystems Dynamics Research, RIKEN, 6-2-3 Furuedai, Suita 565-0874, Japan; chikara.furusawa@riken.jp; 3Universal Biology Institute, The University of Tokyo, 7-3-1 Hongo, Tokyo 113-0033, Japan

**Keywords:** antimicrobial resistance, laboratory evolution, collateral sensitivity

## Abstract

Laboratory evolution studies, particularly with *Escherichia coli*, have yielded invaluable insights into the mechanisms of antimicrobial resistance (AMR). Recent investigations have illuminated that, with repetitive antibiotic exposures, bacterial populations will adapt and eventually become tolerant and resistant to the drugs. Through intensive analyses, these inquiries have unveiled instances of convergent evolution across diverse antibiotics, the pleiotropic effects of resistance mutations, and the role played by loss-of-function mutations in the evolutionary landscape. Moreover, a quantitative analysis of multidrug combinations has shed light on collateral sensitivity, revealing specific drug combinations capable of suppressing the acquisition of resistance. This review article introduces the methodologies employed in the laboratory evolution of AMR in bacteria and presents recent discoveries concerning AMR mechanisms derived from laboratory evolution. Additionally, the review outlines the application of laboratory evolution in endeavors to formulate rational treatment strategies.

## 1. Introduction

Antibiotic overuse in medical and agricultural domains precipitates the rapid emergence of antimicrobial resistance (AMR) and multidrug-resistant bacterial “superbugs” globally. AMR has been considered one of the top 10 global public health threats by the World Health Organization (WHO) [1,2,3]. In 2019, *Escherichia coli*, followed by *Staphylococcus aureus*, *Klebsiella pneumoniae*, *Streptococcus pneumoniae*, *Acinetobacter baumannii*, and *Pseudomonas aeruginosa*, accounted for approximately 1 million deaths attributable to AMR and 3.6 million deaths associated with AMR worldwide [3]. AMR occurs naturally through genetic changes, either via de novo mutations or the acquisition of resistance genes through horizontal gene transfer [4]. Cells resistant to antibiotics exhibit an elevation in the minimal inhibitory concentrations (MICs), which denotes the lowest antibiotic concentration needed to hinder bacterial replication. In contrast, antibiotic persister cells and tolerant cells demonstrate enhanced survival under antibiotic exposure without a concurrent rise in the MIC [5]. Tolerance is the population-level ability to survive a longer duration of transient drug treatment, while persistence is the subpopulation-level tolerance phenomenon [5]. In some cases, a subpopulation among susceptible cells exhibits higher MICs, a phenomenon distinct from persistence, defined as heteroresistance [6,7]. Similar to the resistance developments, higher tolerance, persistence, and heteroresistance abilities can be acquired through genetic mutations selected over time with antibiotic doses [8,9]. Given that both resistance and tolerance contribute significantly to the failure of antibiotic treatments, understanding the mechanisms of their evolution becomes imperative.

Investigations into mutations conferring AMR and persistence have been conducted through laboratory evolution and genome analysis of clinically resistant isolates. These studies have unveiled the causal relationship between mutations and the acquisition of resistance, including target modification, drug inactivation, and drug transport [10,11,12,13]. However, both clinical isolates and laboratory-evolved strains often carry mutations in genes beyond those classified into the three broad categories [14]. These results imply a complex relationship between genetic changes and AMR or persistence, owing to the multifaceted effects of antibiotics on the biomolecular network. To unravel the complexity of AMR or persistence mechanisms and evolution, the quantitative analysis of phenotypic and genotypic changes faces limitations when using clinical isolates due to the absence of nearest ancestral strains and the presence of numerous neutral mutations. Additionally, mutations conferring AMR, which are infrequently identified, are known to be challenging to detect through genome analysis of clinical isolates [14]. Consequently, to overcome these limitations, laboratory evolution, followed by whole-genome sequencing and phenotyping assays, has emerged as a promising methodology.

In light of the substantial upfront investment and prolonged clinical trials required for the development of novel drugs, an alternative approach to combat AMR is urgently needed. Recognizing that the emergence of AMR is rooted in evolutionary dynamics, the application of evolutionary trade-offs to suppress further resistance emerges as a promising strategy [15,16,17,18]. The development of AMR against a primary drug may lead to a dual outcome, inducing either heightened or reduced susceptibility to other drugs concurrently. This phenomenon is known as collateral sensitivity, wherein resistance to a particular drug increases susceptibility to another, or cross-resistance, wherein resistance to one drug amplifies resistance to another (Figure 1) [19]. Antibiotic pairs exhibiting collateral sensitivities and their genetic determinants have been extensively explored through laboratory evolution [20,21,22,23,24,25,26,27]. These studies have identified new collateral-sensitivity and cross-resistance interactions, proposing novel, rational treatment strategies to exploit collateral sensitivity. This review article elucidates the laboratory evolution methodologies employed to comprehend AMR mechanisms and evolution in pathogenic bacteria and presents successful examples of studies on designing rational treatment strategies.

The “landscape” metaphor in evolutionary biology refers to a visual representation of how different traits or characteristics of organisms (in this case, microbial resistance to antibiotics) change over time [28,29]. It is dynamic because the conditions that microbes face, such as exposure to antibiotics or other environmental factors, are constantly shifting. Just as a landscape can have peaks and valleys, the dynamic landscape of resistance illustrates how microbial traits might rise or fall in response to various selection pressures.

The concept of the “dimensionality of phenotypic states” in the context of AMR evolution discusses how traits or characteristics of microbes (like their ability to resist drugs) can be expressed in different ways [30,31]. This entails exploring how these traits can change and adapt over time in response to factors like drug exposure. There are numerous components and features in microbial cells, often measured as omics data such as transcriptome (the set of all RNA transcripts), proteome (the entire set of proteins), metabolome (the complete set of intracellular small-molecule chemicals), and genomic mutations (the loci and the type of mutations) [22,27,32,33,34]. These cellular components contribute to a large number of degrees of freedom, forming high-dimensional data. Despite this complexity, changes related to adaptation and evolution were observed as being effectively limited to a lower-dimensional subspace [22,27]. This implies that, even though there are many variables, only a few of them are crucial for expressing traits that impact the microbe’s ability to evolve and adapt, especially in the context of AMR. Therefore, certain traits related to AMR evolution can be represented or changed in simpler, more predictable ways during evolution by focusing on a few key aspects rather than the entire complexity of the microbe’s characteristics. Simplifying this high-dimensional data into a lower-dimensional representation helps to bridge the gap between the intricate details of cellular components and the essential biological information, such as growth rate, adaptability, and survival. The dimension reduction implies that phenotypic changes are constrained to a lower-dimensional subspace, meaning that certain key traits change more significantly than others [30,31,35]. Once these constraints are acquired through evolution, subsequent adaptations to new environments tend to follow the same restricted paths [35]. These concepts are crucial for understanding how microbes adapt and evolve in response to challenges like antibiotic exposure. The focus on lower-dimensional representations helps to identify key factors that play major roles in adaptation, making it easier to predict and potentially control the evolutionary dynamics of AMR.

## 2. Methodologies of Laboratory Evolution

Analyzing the mechanisms underlying antibiotic resistance and/or tolerance can be accomplished through laboratory evolution, employing a serial transfer approach in the presence of high antibiotic concentrations. These methods involve regularly transferring a portion of the culture to a fresh medium at specific intervals, fostering drug resistance or tolerance (Figure 2A,B). To assess the reliability and stochastic nature of evolutionary changes, multiple independent culture lines are typically run concurrently for each experiment. In experiments of this nature, enhancing the selection pressure during laboratory evolution often involves incrementally raising the antibiotic concentration alongside the development of antibiotic resistance in the evolving cells (Figure 2A). For instance, Lázár et al. conducted laboratory evolution of *E. coli* under 12 antibiotics, progressively elevating the dosage (by 1.5-fold) at every fourth transfer [21]. An alternative approach involves culturing cells in media containing drug gradients, where cells capable of growth at the highest drug concentrations are transferred to subsequent passages (Figure 2B) [22,27,36]. In previous studies, we conducted high-throughput laboratory evolution of *E. coli* using a 2-fold drug gradient across 10 dilutions [22] or a 2^0.25^-fold gradient across 22 dilutions [27]. Compared to the method where the drug dosage is gradually increased at fixed intervals, using drug gradients enables a broader range of selection pressure, providing a more nuanced selection process. This favors cells that exhibit adaptive responses to a range of drug concentrations, allowing for a continuous and more natural selection process. These serial transfer experiments are often laborious due to the requirements of numerous culture conditions and independent culture lines as replicates for the evaluation of reproducibility. Therefore, these experiments are frequently conducted using laboratory automation systems, such as automatic pipetting robots equipped with robot arms, which are capable of precisely aspirating and dispensing sample liquids into microplates and tubes. Some of the robots are equipped with robot arms to transfer plates to equipment used to monitor cell growth, like microplate readers and shaker incubators [37]. It is essential to note that the influence of selection pressure on AMR evolution was assessed by both the drug increment approach and drug gradient approach [36,38]. The drug increment approach suggested that rapidly changing environments limit mutational opportunities by reducing population size and eliminating specific sets of mutations as viable evolutionary options [38]. The findings emphasize the importance of understanding how the rate of environmental change affects both demographic and genetic aspects of evolutionary rescue, especially in the context of unprecedented anthropogenic environmental alterations [38]. On the other hand, the drug gradient approach found that bacterial populations evolving under strong selection developed high levels of cross-resistance against multiple antibiotics, while those evolving under milder selection exhibited weaker cross-resistance [36]. This study also observed that strongly selected populations acquired a higher number of mutations in genes specific to the target pathways of the drugs used for selection [36]. These mutations were more diverse in strongly selected populations compared to those under mild selection [36]. Strongly selected populations often acquired mutations in essential genes with higher fitness costs, while mildly selected populations tended to acquire mutations in genes with lower fitness costs [36].

Instead of employing the serial transfer approach, continuous culturing is also applicable to the laboratory evolution of drug resistance (Figure 2D). Toprak et al. devised the morbidostat system, a tool capable of constantly monitoring cell growth and adjusting drug concentrations within the culture vials to maintain the selection pressure by using feedback control [39]. This system comprises several components, including a flat-bottom glass vial, a magnetic stirrer to mix the culture, an optical density (OD) detection system, and a computer-controlled set of peristaltic pumps used for liquid transfer [39,41]. The distinctive advantage of the morbidostat lies in its ability to dynamically adapt antibiotic concentrations within culture vials, ensuring a sustained drug-induced inhibition, thereby fine-tuning the selection pressure according to the evolving adaptation rate [41]. Moreover, the morbidostat keeps the bacterial population at low densities, preventing nutrient limitations in growth, and modulates the growth rate to align with the fixed dilution rate by adjusting the antibiotic concentration [41]. However, while continuous culturing offers significant advantages, it may not be as suitable for high-throughput experiments in contrast to serial transfer methods using automated systems, as it is constrained by the limited amount of equipment available. Furthermore, the continuous culturing process might promote the formation of biofilms during laboratory evolution, potentially necessitating the replacement of culture vials to avoid biofilm formation [42].

For laboratory evolution toward drug persistence or tolerance acquisition, bacterial populations are intermittently exposed to higher concentrations of antibiotics than MIC (Figure 2C). A typical method is repeating batch culturing between antibiotic treatment and the regrowth cycle [43,44,45]. To select tolerant or persister cells, stationary cultures are treated with antibiotics at 100–200-fold MIC for certain durations (usually ~5 h) [45]. By conducting such cycling experiments, Van den Bergh et al. obtained 40 evolved *E. coli* strains showing 1000- to 10,000-fold increases in persistence against aminoglycoside antibiotics [45]. The increased persistence and/or tolerance of laboratory-evolved cells were also observed in *S. aureus* [46] and other ESKAPE pathogens (*Enterococcus faecium*, *S. aureus*, *K. pneumoniae*, *A. baumannii*, *P. aeruginosa*, and *Enterobacter* species) [47], indicating the generality of the cyclic antibiotic treatment protocols in bacterial acquisition of antibiotic persistence and tolerance.

The colony transfer method serves as an alternative experimental setup for cells forming severe aggregates in liquid media (Figure 2E). For example, *Mycobacterium* forming aggregates in anti-tuberculosis drugs underwent laboratory evolution on agar plates with drug gradients [40]. In this experiment, resistant populations were repeatedly selected from cells at the border of the growth inhibitory zone, created by a filter paper soaked with a drug solution [40]. Microbial evolutionary dynamics towards antibiotics can be directly visualized using soft agar, allowing bacteria to swim and spread (Figure 2F). The microbial evolution and growth arena (MEGA)-plate, consisting of successive areas with different concentrations of antibiotics overlaid by thin, soft agar, was developed [23]. Time-lapse imaging of the MEGA-plate captures real-time processes of mutations and selections in the presence of high dosages of antibiotics. The MEGA-plate stands as a valuable tool for directly visualizing and understanding evolutionary dynamics in bacterial populations [23].

## 3. Insights into AMR Mechanisms from Laboratory Evolution

Laboratory evolution studies provide invaluable insights into the dynamic landscape of resistance, revealing intricate details about the evolutionary trajectories, population dynamics, and metabolic adaptations shaping bacterial responses to drug exposure. The complex interplay of cross-resistance and collateral sensitivity in the evolution of AMR has been intensively analyzed using *E. coli* as a model bacterium [21,22,27]. These studies employ a comprehensive approach, integrating laboratory evolution and genomic sequencing to map cross-resistance and collateral-sensitivity interactions in *E. coli*.

### 3.1. Genotype-Phenotype Relationships in AMR Evolution

Lázár et al. conducted laboratory evolution of *E. coli* by exposing cells to increasing dosages of 12 antibiotics [21]. This study revealed convergent molecular evolution across antibiotics, mutations conferring resistance enhancing sensitivity to other drugs, and the identification of chemogenomic profile similarity as a predictor of cross-resistance [21]. Similar results were observed in other laboratory evolution studies [22,27]. Despite diverse initial conditions, similar mutations and resistance mechanisms emerged under antibiotic selection. Mutations that confer resistance to a specific antibiotic often simultaneously enhance sensitivity to many other drugs, suggesting that resistance evolution is driven by mutations with broad consequence [21,22,27]. Approximately 27% of the observed mutations resulted in proteins with compromised or no activities, highlighting the role of loss-of-function mutations in antibiotic adaptation [21]. Cross-resistance patterns are strongly influenced by chemogenomic profile similarity between antibiotics, indicating that exposure to a single antibiotic often leads to multidrug resistance. Interestingly, cross-resistance between two antibiotics is largely independent of whether they show synergistic effects in combination [21].

To explore the complex relationship between genotype and phenotype in antibiotic-resistant *E. coli* strains, Suzuki et al. also conducted laboratory evolution of *E. coli’s* resistance to 11 antibiotics using a serial transfer method by selecting a population showing the growth under the highest drug concentration possible [22]. Through 90-day laboratory evolution and integrated transcriptome–genome analyses, the study revealed that resistance acquisition to one drug significantly alters susceptibility to others. They demonstrated that a simple linear model can quantitatively predict the resistance of 25 drugs via the expression changes of only eight genes [22]. The eight genes were *acrB*, encoding a subunit of a multi-drug efflux pump; *ompF*, encoding an outer membrane porin protein; *cyoC*, encoding the subunit of cytochrome *bo*_3_ terminal oxidase; *pps*, encoding the phosphoenolpyruvate synthase; *tsx*, encoding a nucleoside channel, a receptor of phage T6 and colicin K; *oppA*, encoding an oligopeptide transporter; *folA*, encoding the dihydrofolate reductase; and *pntB*, encoding the beta subunit of pyridine nucleotide transhydrogenase [22].

To understand the constraints shaping the evolution of AMR, a systematic investigation of evolutionary constraints through high-throughput laboratory evolution of *E. coli* was conducted with the addition of 95 antibacterial chemicals covering a wide range of action mechanisms [27]. Machine learning techniques were employed to analyze phenotype–genotype data, revealing low-dimensional phenotypic states and trade-off relationships associated with drug resistance. The findings provide insights into the intricate interplay between genomic, transcriptomic, and resistance profiles, contributing to a comprehensive understanding of evolutionary constraints in adaptive evolution [27]. Since *E. coli*’s evolutionary dynamics were attributable to a relatively small number of intracellular states, this indicates that *E. coli* is equipped with only a limited number of strategies for antibiotic resistance [27]. This study also quantified collateral responses to other antibiotics and found that 336 and 157 pairs of drugs within the 2162 combinations exhibited cross-resistance and collateral sensitivity, respectively [27]. Heteroresistance, a prevalent phenotype in clinically isolated strains to date [7,48], was observed in 15 out of 33 pairs in the laboratory-evolved strains, suggesting its frequent occurrence in the process of acquiring resistance [27]. It was suggested that the perturbation of metabolic activity by metabolic inhibitors, reactive oxygen species generation, and alteration of cytoplasmic membrane permeabilization by cationic peptides could serve as possible strategies to suppress antibiotic resistance [27].

Laboratory evolution also uncovered the often-overlooked area of metabolic adaptations in response to antibiotic treatment, shedding light on how alterations in cellular metabolism can contribute to antibiotic resistance [14]. This study emphasized the importance of population-level analyses in understanding the evolutionary landscape in response to drug treatment. This study aimed to maximize metabolic adaptation rather than growth adaptation in laboratory evolution protocols [14]. This shift in dynamics allowed for a more comprehensive view of antibiotic-specific metabolic variants, revealing underappreciated genes related to central carbon and energy metabolism [14]. Mutations in metabolic genes that arose in response to antibiotic treatment were found in multiple independent populations and responses to different drugs [14]. Particularly, a mutation in the SucA enzyme, part of the 2-oxoglutarate dehydrogenase complex, increased antibiotic resistance by preventing the antibiotic-mediated induction of tricarboxylic acid (TCA) cycle activity, avoiding metabolic toxicity, and minimizing lethality [14]. These metabolic mutations were identified in clinical *E. coli* pathogens at levels comparable to known resistance mutations, indicating their clinical relevance [14]. It should be noted that TCA cycle activity is known to influence bacterial susceptibility to antibiotics, and its modulation has been linked to both antibiotic resistance and tolerance. The inactivation of the TCA cycle resulted in enhancing persister cell formation in *S. aureus* [49], and down-regulation of the TCA cycle was related to levofloxacin resistance in *Vibrio alginolyticus* [50]. In addition, increasing the TCA cycle flux can promote aminoglycoside uptake, thereby eliminating the drug-resistant Gram-negative bacteria [51,52].

### 3.2. Identified Key Genes Conferring Cross-Resistance and Collateral Sensitivity in E. coli

Our comprehensive high-throughput laboratory evolution of *E. coli* systematically investigated the underlying mechanisms of cross-resistance and collateral sensitivity [27]. The key genes associated with these phenomena in *E. coli* are cataloged in Table 1, drawing upon insights from our previous study. Notably, the study emphasized the pivotal role of mutations in genes governing transporters and porins in mediating antibiotic resistance in *E. coli* [27]. Perturbations in uptake and efflux activities emerged as principal mechanisms governing cross-genetic resistance and heteroresistance [27]. Specifically, the study illuminated the significance of the overexpression of efflux pumps AcrAB/TolC and EmrAB/TolC, coupled with the inactivation of their repressors, in conferring resistance to a spectrum of antibiotics [27]. Furthermore, the investigation pinpointed the involvement of YcbZ, a putative protease implicated in translation and ribosome biogenesis, in mediating cross-resistance against multiple antibiotics [27]. Mutations in PrlF, associated with the PrlF-YhaV toxin–antitoxin system, were found to be linked to resistance against specific antibiotics, such as aztreonam and carbenicillin [27]. The observed cross-resistance was partly ascribed to the diminished expression of OmpF, underscoring the intricate interplay of genes in the stress response [27]. Additionally, the study uncovered collateral sensitivities tied to *prlF*-mediated resistance, revealing trade-offs between acquired resistances and susceptibility to specific drugs, including rifampicin [27]. It was postulated that these collateral sensitivities might be linked to the global mRNA destabilization effect induced by increased RNase activity of PrlF [27]. Moreover, the derepression of an alternative sigma factor, RpoS, resulting from a mutation in the regular RssB, conferred both cross-resistance and collateral sensitivity to various drugs [27]. The proposed mechanism for such collateral sensitivity involved the competition between RpoS and the housekeeping sigma factor RpoD, leading to decreased carbon source availabilities and diminished competitiveness for low concentrations of nutrients [27]. The subsequent Section 3.5 elaborates on the trade-off mechanism of aminoglycoside resistance and drugs unrelated to aminoglycosides.

### 3.3. Costs of AMR Evolution

The impact of evolution in the absence of antibiotics on the fitness effects of resistance mutations in *E. coli* was analyzed [53]. Rifampicin-resistant and -sensitive *E. coli* were subjected to experimental evolution in a drug-free environment [53]. The fitness effects of newly acquired resistance elements were quantified under antibiotic-free conditions [53]. The results revealed that streptomycin-resistance mutations exhibited smaller fitness effects in rifampicin-resistant genotypes adapted to antibiotic-free growth medium compared to non-adapted genotypes [53]. This epistatic variation in the costs of resistance was observed not only between different resistance mutations, but also between resistance elements and beneficial mutations acquired during adaptation to drug-free conditions [53].

The rates of resistance evolution in bacteria exposed to different antibiotics were analyzed by quantifying the distribution of fitness effects (DFE) of mutations [54]. The DFE of mutations refers to the range and characteristics of fitness changes that result from genetic mutations within a population. The DFE is thought to be a crucial factor influencing the evolutionary dynamics of drug resistance in bacterial populations [54]. The DFE in the presence of eight antibiotics, representing various modes of action, was comprehensively measured [54]. Surprisingly, the width of the DFE varied dramatically between antibiotics, with some drugs exhibiting a lower DFE width than that in the absence of stress [54]. This divergence in DFE width was attributed to distinct, drug-specific dose–response characteristics [54]. This research identified resistance variability and dose sensitivity as key drug-specific properties shaping the DFE [54]. Nitrofurantoin, in particular, exhibited exceptionally low resistance variability, limiting the evolution of resistance. Laboratory evolution experiments using nitrofurantoin confirmed slow resistance evolution via reproducible mutations [54].

Metabolic constraints on AMR evolution were investigated through laboratory evolution of *E. coli* under three different antibiotics (ampicillin, chloramphenicol, and norfloxacin) while growing on glucose or acetate as the sole carbon source [55]. Profiling more than 500 intracellular and extracellular metabolites in 190 evolved populations revealed that carbon and energy metabolism strongly constrain the evolutionary trajectories of antibiotic resistance [55]. Both the speed and mode of resistance acquisition were influenced by metabolic adaptations [55]. This study demonstrated that resistance evolves more rapidly on glucose compared to acetate, indicating greater metabolic plasticity during respiro-fermentative metabolism [55]. Environmental conditions played a crucial role in determining the trade-off between the costs and benefits of resistance mutations, influencing how rapidly a resistant mutant establishes itself within the population. This study identified condition-dependent compensatory mechanisms in antibiotic-resistant populations, including shifts in metabolic pathways [55]. For example, there was a shift from respiratory to fermentative metabolism of glucose upon overexpression of efflux pumps, highlighting the role of metabolic adaptations in response to antibiotic selection pressure [55]. Metabolome-based predictions revealed emerging weaknesses in antibiotic-resistant strains, such as the hypersensitivity to fosfomycin in ampicillin-resistant strains [55]. Additionally, monitoring ~750 intracellular metabolites in *E. coli* immediately after antibiotic exposure revealed the early bacterial response to antimicrobial compounds and potential weaknesses in terms of tolerating antibiotic therapies [56]. For example, unbalanced ammonia metabolism contributed to increased chloramphenicol toxicity, linking it to the generation of reactive oxygen species [56].

The transferability of ten resistance-conferring mutations and resistance genes (*acrR*, *envZ*, *fis*, *gyrA*, *mprA*, *ompC*, *phoQ*, *soxR*, *trkH*, and *ycbZ*) between *E. coli* and the closely related *S. enterica* subsp. serovar Typhimurium strain was investigated [57]. This study revealed that the resistance mutations in *E. coli* often did not confer resistance in *S. enterica* [57]. Surprisingly, in some cases, these mutations led to drug hypersensitivity in *S. enterica* [57]. In-depth analysis of a key gene involved in aminoglycoside resistance (*trkH*) indicated that extreme differences in mutational effects between the two species were attributed to preexisting mutations in other genes [57]. This study emphasized the limited conservation of mutational effects driving collateral sensitivity and cross-resistance phenotypes among antibiotic-resistant bacteria. The effects strongly depended on the genetic background, and even a single resistance mutation could alter collateral responses to other antibiotics.

### 3.4. Impact of Multidrug Combinations on AMR Evolution

The impact of multidrug combinations on the competitive selection between sensitive and resistant bacterial populations was analyzed using doxycycline-resistant and doxycycline-sensitive *E. coli* [58]. In a hyper-antagonistic class of drug combinations, a drug can render the combined treatment selective against its resistance allele. This inverted selection is insensitive to the underlying resistance mechanism and occurs at sublethal concentrations while maintaining inhibition of the wild type. The findings suggest a trade-off between the effect of drug interactions on absolute potency and the relative competitive selection imposed on emerging resistant populations. The study emphasizes a previously unappreciated feature of the fitness landscape for the evolution of resistance and points to potential avenues for designing antimicrobial combinations with improved selection against resistance [58].

The dynamics of resistance development in *E. coli* when exposed to five different single antibiotics (ciprofloxacin, amikacin, tetracycline, chloramphenicol, and piperacillin) and all possible antibiotic drug pairs were investigated [59]. The degree of resistance development against drug combinations was found to be linked to collateral sensitivity and the resistance that occurred during adaptation to the component drugs [59]. It was confirmed that drug resistance causing collateral sensitivity was suppressed in the growth environment under the treatment of amikacin and the other drug pairs [59]. This implies that drug combinations can limit the evolution of resistance if resistance to one drug in the combination results in collateral sensitivity to another drug.

The evolutionary conservation of collateral responses to five clinically relevant antibiotics (cefepime, ciprofloxacin, gentamicin, meropenem, and tetracycline) across species, particularly within the ESKAPE pathogens, was investigated [60]. This study identified 14 instances of universal cross-resistance that accounted for 80% of all the collateral responses and two global collateral sensitivity relationships (cefepime and gentamicin pair and ciprofloxacin and gentamicin pair) among the evolved lineages [60]. Although the genetic basis for the collateral sensitivity was unclear, genomic analyses revealed divergent and conserved evolutionary trajectories, suggesting that collateral responses may be preserved across species for the selected drugs [60].

### 3.5. Mechanism of the Trade-Off between AMR Evolution

The trade-off relationships between aminoglycoside resistance and other drugs were demonstrated by several studies (Figure 3) [20,22]. Mutations in genes whose products are involved in respiration, such as cytochrome *bo*_3_ oxidase and NDH-I, were commonly identified in aminoglycoside-evolved strains [20,22]. Since aminoglycoside uptake is known to require proton motive force (PMF) [61,62], decreased respiration results in a reduced PMF and inhibition of aminoglycoside uptake. On the other hand, activation of the multidrug efflux pump AcrAB/TolC through the inactivation of its repressor AcrR is a common multidrug resistance mechanism against various antibiotics, except for aminoglycosides [21,22,27]. Since AcrAB/TolC is a PMF-dependent proton antiporter [63,64], a decrease in PMF leads to hyper-susceptibility to other drugs that are discharged by AcrAB/TolC. Therefore, the acquisition of simultaneous resistances against both aminoglycoside and other drugs that are discharged by AcrAB/TolC poses a significant challenge. This difficulty was experimentally demonstrated by the laboratory evolution of *E. coli* under the simultaneous addition of two-drug combinations [59,65].

### 3.6. Population Dynamics of AMR Evolution

The population dynamics of norfloxacin resistance were analyzed in a bioreactor [66]. The majority of individual evolved isolates were less resistant, with lower MICs than the group MIC. Although highly resistant isolates with higher MICs than the bioreactor concentration were rare, their presence was notable [66]. The highly resistant mutants were often present in low abundance in the population, and their emergence preceded increases in the group MIC [66]. The beneficial effect of the highly resistant mutants on the major lower resistant mutants resulted from indole production [66]. The highly resistant mutants endured a fitness cost to produce and share indole, acting as a form of altruism [66]. The production of indole by highly resistant mutants served to turn on drug efflux pumps and oxidative-stress-protective mechanisms, enhancing the survival of less resistant isolates [66]. This altruistic behavior, despite imposing a fitness cost on the highly resistant mutants, allowed weaker constituents to survive antibiotic stress [66].

### 3.7. AMR Evolution in Spatially Structured Environments

The spatial dynamics of bacterial evolution, emphasizing the interplay between mutational diversity, spatial constraints, and the ultimate fitness of the evolving population, were investigated using a growth arena (MEGA)-plate, a large experimental device allowing for the study of evolution in a large, spatially structured environment [23]. Mutants with high resistance to trimethoprim or ciprofloxacin did not always lead the evolutionary front. Highly resistant mutants could be trapped behind more sensitive lineages [23]. The physical blocking of mutants by each other was observed, reminiscent of phenomena observed in biofilm formation [23]. Mutations that increased resistance often came at the cost of reduced growth, which was restored by compensatory mutations [23]. Compensatory mutants were spatially restricted, appearing in localized spots behind the evolving front [23]. These mutants, though able to outcompete their parent in certain conditions, were constrained from contributing to the ultimate evolutionary course of the population when spatially restricted [23]. The fitness of the bacterial population was found to be determined not only by the fittest mutants, but also by those that were sufficiently fit and arose sufficiently close to the advancing front [23]. Mutants with enhanced resistance and compensatory mutations, when appearing at the front without being physically blocked, accelerated the adaptive process [23].

### 3.8. Fitness Landscape and AMR Evolution

The fitness landscape represents the relationship between genotype or phenotype and fitness in the context of antibiotic resistance evolution [67]. Since the fitness landscape is crucial for explaining and predicting evolutionary trajectories, the phenotype-based fitness landscape for AMR evolution in *E. coli* was investigated by quantifying the multidimensional phenotypic changes, i.e., time-series data of resistance for eight different drugs [67]. Because of the challenge regarding the high dimensionality of genotypic changes, this study quantified multidimensional phenotypic changes using time-series data on resistance for eight different drugs. This study reveals that different peaks in the phenotype-based fitness landscape correspond to different drug resistance mechanisms [67]. This finding supports the validity of the inferred landscape as a representation of the relationship between phenotype and fitness in the context of AMR [67]. A distinctive aspect of the study is the empirical approach used to infer the fitness landscape based on antibiotic resistance profiles rather than genotypes. The immense number of possible genotype changes associated with AMR makes empirical fitness landscapes based on genotypes less capable of predicting and controlling evolution. This study demonstrates that the directions of evolution predicted by the inferred phenotype-based fitness landscape align with the observed experimental trajectories, at least for evolution under certain antibiotics (tetracycline, kanamycin, and norfloxacin) [67]. This consistency suggests that resistance profiles capture the internal degrees of freedom of *E. coli* for predicting evolution.

### 3.9. Antibiotic Tolerance and Persistence Development

It has been demonstrated that conditions inducing bacterial tolerance can operate in various non-specific ways, both in vitro and during infection [68,69,70,71,72]. While tolerant populations exhibit high survival rates under antibiotic treatment, they suffer from impaired proliferation during infection. In contrast, antibiotic persistence is highlighted as a risk-limiting strategy, enabling bacteria to survive antibiotic treatment without compromising their ability to colonize the host [68]. The findings emphasize the importance of understanding the dynamics of these recalcitrance mechanisms to improve the design of more effective antibiotic therapies, considering the different fitness trade-offs associated with tolerance and persistence.

Metabolic homeostasis, the balance of metabolic processes in the cell, can be perturbed to promote antibiotic persistence [73,74,75,76,77,78,79,80,81]. A laboratory evolution and population-wide sequencing identified mutations in respiratory complex I (type I NADH dehydrogenase; NDH-I) as key contributors to increased persister formation in *E. coli* [45]. This finding was consistent across both model and pathogenic *E. coli* strains [45]. Mutations in NDH-I that compromise proton pumping led to significant cytoplasmic acidification upon metabolic perturbations [45]. The proposed mechanistic model suggests that strong metabolic perturbations, such as entering the stationary phase or abrupt nutrient shifts, result in cytoplasmic acidification [45]. Mutations in NDH-I exacerbate this acidification, acting as central signaling hubs connecting perturbed metabolic homeostasis with persister cell formation [45].

## 4. Designing Rational Treatment Strategies through Laboratory Evolution

Addressing the escalating threat of AMR, there is a growing interest in combination-based treatments to boost efficacy. One strategy involves identifying collateral sensitivities. It was demonstrated, through laboratory evolution and whole-genome sequencing, that alternating drugs during treatment effectively slows resistance evolution compared to single-drug treatments, exploiting evolutionary trade-offs [16]. Melnikov et al. reported that amplifying the fitness cost of drug resistance, particularly in tavaborole-resistant *E. coli*, fosters natural competition between resistant and susceptible cells, showcasing collateral sensitivity [82]. Tavaborol, a novel, synthetic small-molecule inhibitor of protein synthesis, received approval from the Food and Drug Administration in 2014 for the treatment of onychomycosis (nail fungus) [83,84,85,86]. Tavaborole functions by targeting the editing domain of leucyl-tRNA synthetase (LeuRS). Through covalent binding with tRNALeu, tavaborole prevents the dissociation of tRNALeu from LeuRS [84]. It has been demonstrated that tavabolore and its deliberatives are effective against multidrug-resistant bacterial pathogens, including *E. coli*, *P. aeruginosa* [86], *S. aureus* [87], *Mycobacterium tuberculosis* [88,89,90,91], and *S. pneumoniae* [92]. The resistant mechanism against tavaborole was elucidated through the acquisition of a specific mutation in the LeuRS editing domain, as revealed by whole-genome sequencing of clinical isolates of *E. coli* [93] and *P. aeruginosa* [94]. Since the tavabole-resistant *E. coli* strain carrying the LeuRS mutation showed hypersensitivity to norvaline (a chemical analog of leucine), the cost of tavabole resistance can be amplified in the presence of norvaline, which is misused for protein synthesis by the resistant cells [82]. Indeed, Melnikov et al. demonstrated that simultaneous tavaborole and norvaline treatment slowed tavaborole resistance evolution [82]. It is crucial to note that complete suppression was not achieved, indicating a potential risk of multidrug resistance [82].

### 4.1. Collateral Sensitivity as a Potential Strategy for Designing Rational Antibiotic Treatment

The degree to which a drug combination can reduce the evolution of resistance depends on the interplay between collateral resistance and collateral sensitivity in its component drugs. If resistance to one drug leads to collateral sensitivity to another drug, the combination is more effective in limiting the evolution of resistance [59]. The impact of drug combination therapy on AMR in *E. coli* was investigated through laboratory evolution [65]. This study revealed that certain combinations of drugs (aminoglycoside amikacin and chloramphenicol or enoxacin) with collateral sensitivity could suppress the acquisition of resistance [65]. The effectiveness of drug combination with collateral sensitivity in an opportunistic human pathogen *P. aeruginosa* was also investigated [24]. Unlike patterns observed in other bacterial species, collateral effects in *P. aeruginosa* showed interspecific differences in evolutionary trade-offs [24]. Interestingly, diverse patterns of collateral sensitivity and cross-resistance have emerged among replicate populations adapted to the same drug [24]. Genomic analysis of evolved populations reveals distinct evolutionary paths, determining whether bacteria become cross-resistant or collaterally sensitive [24]. The study identified key regulatory genes (*nalC*, *mexZ*, and *pmrB*) associated with collateral effects, providing insights into the mechanisms governing these responses [24]. Overall, these studies contributed valuable insights into the development of novel antibiotic therapies that leverage fitness trade-offs during drug resistance evolution.

### 4.2. Collateral Sensitivity Cycling

At the outset, antibiotic cycling was dependent on the inherent fitness cost of AMR. A higher fitness cost meant a more rapid decrease in the resistance frequency within a population when the selective pressure from drug treatment was absent [95]. Nevertheless, the stability of fitness costs linked to resistance is not guaranteed, as compensatory mutations that diminish these costs can compromise the effectiveness of cycling [96,97]. Therefore, using a collateral sensitivity network based on the data set of *E. coli* AMR evolution to 23 clinically used drugs, a new treatment framework, collateral sensitivity cycling was proposed (Table 2) [15]. This approach involves using drugs with compatible collateral sensitivity profiles sequentially to treat infections, selecting against drug resistance development [15]. The study identified numerous drug sets, demonstrating that cyclic deployment of antibiotics like gentamicin and cefuroxime could effectively counter resistance [15]. Collateral sensitivity cycling, based on reciprocal collateral sensitivities, differs fundamentally from current drug cycling methods, offering a sustainable treatment paradigm [15]. The findings suggest its potential application in managing Gram-negative bacterial infections and emphasize its relevance for chronic infections and cases involving multiple pathogens [15]. Clinical studies are needed in order to validate these principles in hospital settings. The study anticipates that collateral sensitivity cycling will contribute to the sustainable use of drugs in clinical disease management [15].

Using the morbidostat platform, it was demonstrated that multidrug resistance could be manipulated by administering pairs of antibiotics and cycling between them in an ON/OFF manner [98]. Cyclin use of polymyxin B and other antibiotics completely suppressed the development of resistance to one of the antibiotics [98]. This study also emphasized the importance of exploiting the collateral sensitivity of various antibiotics for designing effective treatment methods that can suppress or reverse drug resistance [98].

The impact of an antibiotic cycling strategy on the prevalence of antibiotic-resistant, Gram-negative bacteria was investigated through a cluster-randomized crossover study conducted across eight intensive care units (ICUs) in Europe [99]. Antibiotic-resistant, Gram-negative bacteria were operationally defined as *Enterobacteriaceae* with extended-spectrum β-lactamase production or resistance to piperacillin–tazobactam, along with *Acinetobacter* spp. and *P. aeruginosa* displaying resistance to piperacillin–tazobactam or carbapenems within the study parameters [99]. The study centered on three antibiotic groups—third-generation or fourth-generation cephalosporins, piperacillin–tazobactam, and carbapenems—and evaluated the prevalence of resistant bacteria during cycling (where a specific antibiotic class was preferentially employed for a 6-week period) and mixing (where the antibiotic class was altered for each consecutive patient) [99]. Despite observed variations in antibiotic utilization across ICUs, the comprehensive analysis of study antibiotics’ overall volume and consumption patterns during cycling and mixing revealed no statistically significant differences [99]. The principal examination, accounting for patients present during monthly point-prevalence surveys, disclosed a mean prevalence of antibiotic-resistant, Gram-negative bacteria of 23% during cycling and 22% during mixing, with no discernible statistical distinction between the two strategies [99]. The study concluded that antibiotic cycling does not manifest a substantial reduction in the prevalence of antibiotic-resistant bacteria in ICUs [99]. It is noteworthy to mention that this study was conceived in 2010 [99], predating the development of collateral sensitivity cycling introduced by Imamovic et al. in 2013 [15]. Consequently, there exists a positive prospect regarding the clinical efficacy of this strategy; however, additional clinical evidence is imperative to adjudicate the reliability of this approach. To implement collateral sensitivity cycling in real-world scenarios, the collateral effects of clinically relevant drug combinations must exhibit robustness across genetically diverse backgrounds. Therefore, the robustness of collateral sensitivity in various clinical resistant strains has been intensively analyzed. For example, the study demonstrated the robustness of the collateral sensitivity of ciprofloxacin towards gentamicin, fosfomycin, ertapenem, and colistin in *E. coli* [100], and towards gentamicin, fosfomycin, colistin, aztreonam, and tobramycin in *P. aeruginosa* [25,101]. The efficacy levels of aminoglycoside and β-lactam in eradicating quinolone-resistant *P. aeruginosa* in patients suffering from cystic fibrosis were also demonstrated [25].

### 4.3. Sequential Drug Regimens Based on Collateral Sensitivities

The potential benefits of sequential drug regimens based on collateral sensitivities, where exposure to a first drug induces susceptibility to a second, were investigated [26]. This study suggests that sequential drug regimens derived from in vitro evolution experiments may have overstated therapeutic benefits [26]. Predictions of collaterally sensitive responses can lead to cross-resistance, where resistance to the first drug also results in resistance to the second [26]. Using mathematical modeling parametrized with combinatorially complete fitness landscapes for *E. coli*, the likelihood of collateral sensitivities being stochastic was quantified [26]. This study also conducted laboratory evolution with *E. coli* under cefotaxime, a beta-lactam antibiotic [26]. The study involved 60 parallel evolutionary replicates to demonstrate the extent of heterogeneity in second-line drug sensitivity [26]. The results showed that a second drug can stochastically exhibit either increased susceptibility or increased resistance when following a first drug [26]. Genetic divergence was identified as the driver of this differential response through targeted and whole-genome sequencing [26]. Collateral sensitivity was found to be rare and not universal [26]. Different mutations, representing different evolutionary trajectories, contributed to heterogeneity in the collateral response. These results emphasized that the success of evolutionarily informed therapies is predicated on a rigorous, probabilistic understanding of the contingencies that arise during the evolution of drug resistance [26]. This study proposed collateral sensitivity likelihoods as critical statistical benchmarks for the clinical translation of sequential drug therapies [26]. The likelihoods were derived through empirical observations and mathematical modeling. These findings provide insights into the challenges and considerations involved in designing effective sequential drug therapies to combat AMR.

### 4.4. Optimization of Antibiotic Treatment for Chronic Infections by Targeting Phenotypic States

In the in vitro AMR evolution process of *P. aeruginosa* in synthetic cystic fibrosis (CF) sputum medium towards 24 clinically relevant antibiotics, a phenotypic convergence towards distinct states associated with specific mutations in antibiotic resistance genes was observed, exhibiting collateral sensitivity to several antibiotic classes [25]. This study suggests that chronic infections could be more effectively treated by targeting these phenotypic states linked with particular mutations [25]. Collateral sensitivity, observed even when the organism is already resistant to certain drugs, can be exploited to design rational treatment strategies. For example, the application of one antibiotic may lead to subsequent resistance development and phenotypic convergence, but this could enhance the action of another antibiotic to which collateral sensitivity has been observed [25]. This study emphasizes the potential clinical impact of collateral sensitivity in optimizing treatment strategies for chronic infections, particularly in conditions like CF, where patients undergo repeated rounds of antibiotics or lifelong therapies [25]. The application of ciprofloxacin for the treatment of *P. aeruginosa* infections was found to lead to subsequent resistance development and phenotypic convergence [25]. Importantly, this resistance could enhance the efficacy of tobramycin due to the collateral sensitivity observed in resistant strains [25]. Similar enhancement was observed for colistin action in bacteria that developed resistance to ciprofloxacin or aztreonam. Mutations in the *nfxB* gene encoding a negative transcriptional regulator of MexCD-OprJ efflux system associated with quinolone resistance [102] were linked to collateral sensitivity across various drug classes and phenotypic convergence [25]. Notably, *nfxB* mutants were eliminated during antibiotic treatment in a CF patient, emphasizing the potential for targeting specific mutations associated with collateral sensitivity in optimizing chronic infection treatment [25]. This study suggests that, with monitoring, *nfxB* gene mutations or MexC protein abundance could serve as biomarkers for collateral sensitivity in clinical settings [25].

### 4.5. Suppression of Tolerance Acquisition by Cycling Antibiotics with Different Metabolic Dependencies

Antibiotics exhibit varying degrees of metabolic dependencies, and those weakly dependent on metabolism maintain effectiveness even when targeting dormant cells [103]. The metabolic dependency of *E. coli* was assessed by examining the bactericidal effectiveness of various antibiotics across a range of nutrient availabilities (Table 2) [103]. This investigation revealed distinct categories of antibiotics regarding their reliance on metabolism, with ampicillin and ciprofloxacin identified as strongly dependent, while gentamicin, halicin, and mitomycin C were categorized as weakly dependent [103]. Notably, ampicillin and ciprofloxacin exhibited diminished bactericidal efficacy when faced with nutrient-depleted cells, in contrast to mitomycin C, which maintained its effectiveness under these conditions [103]. The effectiveness of strongly dependent drugs exhibited a positive correlation with increased intracellular ATP concentration (and, therefore, metabolic state). Conversely, the effectiveness of weakly dependent drugs appeared to be largely unaffected by variations in intracellular ATP concentration [103]. During chronic infections, the pathogen’s metabolism can be downregulated, leading to failure in eradicating pathogens due to antibiotic tolerance [103,104,105,106]. The rate of tolerance evolution was shown to depend on antibiotic metabolism dependencies [107]. Tolerance evolved more readily against antibiotics which were strongly dependent on bacterial metabolism, such as ampicillin and ciprofloxacin, compared to those minimally affected by the metabolic state, such as gentamicin, halicin, and mitomycin C [107]. Additionally, this study demonstrated that cycling antibiotics with different metabolic dependencies can interrupt the evolution of tolerance, extending the treatment efficacy duration [107]. Therefore, cycling strategies alternating between antibiotics which are strongly and weakly dependent on metabolism were proven to be effective in terms of delaying tolerance evolution, offering insights into combination treatments. This study highlights that differences in antibiotic metabolic dependencies could guide the design of customized treatment strategies, balancing concerns of toxicity and tolerance evolution [107].

### 4.6. Long-Term Clearance Efficacy of Drug Combinations

To understand how drug combinations affect bacterial long-term clearance at clinically relevant concentrations, the survival of *S. aureus* during prolonged exposure to pairwise and higher-order cidal drug combinations was systematically quantified [108]. This study found a phenomenon of reciprocal suppression in clearance interactions. Unlike growth inhibition and early killing, the efficacy of drug combinations decreases rather than increases as more drugs are added [108]. This finding challenges conventional wisdom and underscores the need for a nuanced understanding of drug interactions over time. This study suggests that strong suppressive clearance interactions are suggestive of induced persistence in the presence of specific drug combinations [108]. Adding drugs that target non-growing persisters, such as mitomycin C and daptomycin, restored the efficacy of otherwise suppressive drug mixes [108]. The lack of correlation between growth inhibition interactions and long-term clearance interactions cautions against relying solely on growth inhibition phenotypes when prescribing drug combination therapies. Drug combinations with high inhibitory efficacy may not necessarily have increased long-term efficacy. This study also highlights that adding β-lactamase inhibitors, commonly used to potentiate treatment against β-lactam-resistant strains, counterintuitively reduces the long-term clearance efficacy of drug combinations against such strains [108]. This cautions against certain clinically prescribed combinations. Despite the challenges posed by reciprocal suppressive interactions, this study suggests that they open up opportunities for designing treatment regimens that are inherently selective against resistance to any one of their agents [108].

### 4.7. Clinical Evidence Supporting the Efficacy of Antibiotic Combination Therapy Involving Aminoglycosides Is Substantial

The utilization of such combinations has become widespread, particularly in addressing severe hospital-acquired infections caused by multidrug-resistant species, owing to the apparent effectiveness of the evolutionary trade-off relationships between aminoglycoside resistance and other drugs [109,110,111]. A retrospective study focusing on bacteremia predominantly induced by Enterobacter cloacae in cancer patients was conducted by Bodey et al. in 1991 [112]. The study assessed response rates, considering the eradication of all signs and symptoms of Enterobacter infection as the endpoint. The results indicated response rates of 59% and 74% with the single use of aminoglycoside or penicillin, respectively, while simultaneous administration of aminoglycoside and penicillin demonstrated a higher response rate of 78% [112]. Examining the clinical efficacy and safety of combination therapy involving aminoglycoside antibiotic gentamicin and macrolide antibiotic azithromycin for treating urogenital gonorrhea caused by Neisseria gonorrhoeae infection, in a study conducted by Kirkcaldy et al. in 2014, reported promising outcomes [113]. The randomized, multisite, open-label, noncomparative trial revealed a 100% microbiological cure rate among 202 evaluable participants receiving gentamicin/azithromycin [113]. Nevertheless, there exist discrepancies regarding the use of such a two-combination therapy [109]. For instance, a randomized trial comparing β-lactam monotherapy with β-lactam-aminoglycoside combination therapy for sepsis in immunocompetent patients, inclusive of various pathogens such as *S. aureus*, *Enterobacteriaceae*, and *P. aeruginosa*, concluded that the addition of an aminoglycoside to β-lactams is discouraged due to unaltered fatality rates [114]. A subsequent study corroborated this conclusion; however, this study additionally concluded a survival benefit of β-lactam-aminoglycoside combination therapy when prescribed to children with multidrug-resistant Gram-negative bacteria [110].

Despite lingering uncertainty, the use of aminoglycosides in combination therapy has been recommended, emphasizing risk stratification [111,115]. It is crucial to note that prior combination therapies were designed based on the synergistic effects of antibiotic combinations [109]. Recent laboratory evolution experiments, however, have challenged this perspective by demonstrating that the evolvability of AMR remains independent of whether the combinations exhibit synergistic effects [21,59,116,117]. Rather than relying on observed drug interactions, the results revealed a discernible pattern linking genetic trajectories to resistance evolution [116]. Therefore, future endeavors must delve into elucidating the mechanisms of evolutionary constraints and identifying specific drug pairs to inform the design of rational treatment strategies for combination therapy.

### 4.8. Application of Antimicrobial Peptides (AMPs) for Combating AMR

The emergence of drug resistance to conventional antibiotics necessitates the exploration of alternative agents, such as AMPs. AMPs demonstrate broad-spectrum antimicrobial activity characterized by high specificity and low toxicity, derived from both synthetic and natural sources [118]. They play a pivotal role in the host’s innate immunity against various microorganisms, including bacteria, fungi, parasites, and viruses [118]. AMPs are considered promising alternatives due to their capacity to act on multiple targets, combating drug-resistant bacteria [118]. Therefore, current clinical applications of AMPs primarily focus on bacterial infections. Cationic AMPs, effective against bacterial pathogens, target unique anionic components such as LPS in Gram-negative bacteria and lipoteichoic acid in Gram-positive bacteria [118]. The structural diversity of AMPs and their ability to target specific lipids or intracellular proteins contribute to their selectivity against bacterial species or strains. AMPs induce membrane permeation, leading to intracellular content leakage, or penetrate membranes for intracellular effects [118]. In the context of biofilms—microbial aggregates forming on tissues or medical implants—AMPs offer diverse mechanisms to target various biofilm properties [119].

A systematic examination of resistance evolution to 14 diverse cationic AMPs and 12 antibiotics in *E. coli* was conducted [120]. The laboratory evolution of *E. coli* against these AMPs and antibiotics demonstrated that certain AMPs, such as tachyplesin II and R8, completely suppressed resistance acquisition during approximately 120 generations of laboratory evolution, sufficient time to allow for the acquisition of resistances to the 12 antibiotics [120]. Tachyplesins I and II, antimicrobial peptides showing potent activity against various pathogens, exhibited no observed resistance acquisition in a mutator *E. coli* strain or in pathogens including *S. enterica* subsp. serovar Typhimurium, *K. pneumoniae* subsp. *pneumoniae*, and *A. baumannii*, indicating a very low probability of resistance acquisition against tachyplesin II [120]. Increased hydropathicity and fewer polar and positively charged amino acids were associated with reduced resistance during laboratory evolution [120]. In contrast to AMR evolution, no cross-resistance to AMPs or limited horizontal gene transfer for AMP resistance were observed [120]. The possibility of AMP resistance through horizontal gene transfer was assessed by heterologous expression of metagenomic DNA fragments from soil samples (1.8 million clones) in *E. coli* [120].

The effectiveness of combinations of AMPs on AMP evolution in *S. aureus* was also explored [121]. It was demonstrated that treatment with certain AMP combinations delays resistance evolution compared to individual AMPs [121]. The lowest resistance was observed with a random AMP library containing over a million peptides [121]. The study highlighted the correlation between resistance evolution rate, individual AMP resistance cost, and cross-resistance. AMP-resistant strains often remain sensitive to other AMPs, reducing concerns regarding broad-range resistance [121]. Combining AMPs, especially those with high resistance costs, has proven effective in hindering resistance evolution, emphasizing the potential of AMP cocktails for sustainable treatment against antibiotic-resistant pathogens.

**Table 2 antibiotics-13-00094-t002:** Three evolutionary-based strategies to combat AMR.

Strategy	Advantages	Disadvantages	Clinical Trial
Combination of drugs with collateral sensitivity [16,65,82]	Reduce the supply of effective mutations. The trade-off relationship between the drug pair corners pathogens into an evolutionary dead end.	The overall benefits of combinations are not always evident in routine clinical outcomes or from single trials, necessitating a more comprehensive synthesis of clinical data. See Section 4.7.	Positive outcomes have been reported against, e.g., *E. coloacae* [112], *N. gonorrhoear* [113], and multidrug-resistant Gram-negative bacteria [110]. Discrepant results were also reported against various pathogens, including *S. aureus*, *Enterobacteriaceae*, and *P. aeruginosa* [114]
Collateral sensitivity cycling [15,98]	This strategy upgrades the previous drug-cycling strategy dependent on unreliable fitness costs. Instead, this strategy relies on limiting the evolution of drug resistance.	The potential therapeutic advantage might be overemphasized, with genetic divergence identified as the underlying factor influencing diverse responses, leading to either heightened or diminished resistance to subsequent drugs [26].	A European cluster-randomized crossover study determined that the practice of antibiotic cycling does not lead to a reduction in the prevalence of antibiotic-resistant Gram-negative bacteria carriage among patients admitted to intensive care units [99].
Cycling antibiotics with different metabolic dependencies [107]	This strategy can interrupt the evolution of tolerance.	Clinical studies are essential to validate its efficacy in real-world hospital settings.	To date, no relevant clinical trial has been reported yet.

## 5. Conclusions

Laboratory evolution studies revealed the multifaceted nature of AMR evolution, providing crucial insights into its mechanisms. Cross-resistance and collateral sensitivity, influenced by genomic and phenotypic changes, pose challenges and opportunities for developing effective treatment strategies. The integration of laboratory evolution, genomics, and phenomics is crucial for unraveling the complex landscape of AMR, providing valuable insights for future studies and the development of innovative therapeutic approaches.

Convergent molecular evolution across antibiotics was revealed, indicating common resistance mechanisms [21,22,27]. Mutations conferring resistance often simultaneously enhance collateral sensitivity to other drugs, suggesting pleiotropic effects [21,22,27]. Therefore, collateral sensitivity can be exploited to design rational treatment strategies [15,24,25,26,59,65]. Future research is required in order to refine strategies based on collateral sensitivity, understand the heterogeneity in collateral responses, and explore the long-term efficacy of drug combinations. To validate the effectiveness of collateral sensitivity in real-world scenarios, an investigation of the degree to which collateral sensitivity observed in laboratory settings translates to clinical outcomes will be highly required. In addition, the diversity of collateral sensitivity patterns among different bacterial species and strains needs to be explored. It has been suggested that the success of evolutionarily-informed therapies depends on a rigorous probabilistic understanding of contingencies [26]; probabilistic understanding of collateral sensitivities is also important for the success of such evolutionarily-informed therapies.

Recognizing that AMPs and phages serve as viable alternatives to antibiotics, effective combinations should extend beyond antibiotics alone. The design of combinations should not be confined solely to antibiotics. Presently, the development of an ideal drug capable of completely suppressing resistance acquisition remains elusive. Similar to antibiotics, reports of resistance acquisition to AMPs and phages exist [120,122]. The distinct pros and cons associated with antibiotics, AMPs, and phages [123,124] offer endless possibilities for effective combinations. It is important to highlight that the use of aminoglycosides in combinations may pose a risk of notable side effects that potentially surpasses the clinical benefits [109,114]. Consequently, there is a strong need to identify drug pairs that exhibit evolutionary trade-offs, distinct from the reliance on aminoglycosides. To identify ideal combinations based on their evolutionary constraints, further high-throughput laboratory evolution will be necessary to establish a robust starting point.

## Figures and Tables

**Figure 1 antibiotics-13-00094-f001:**
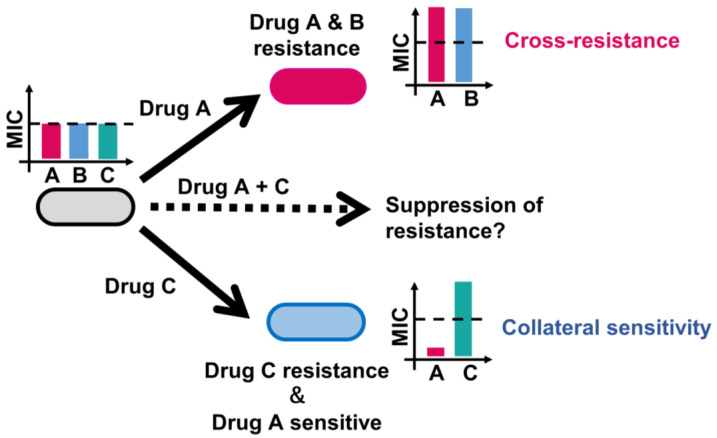
Cross-resistance and collateral sensitivity. Cross-resistance refers to the phenomenon where bacteria, having developed resistance to a drug A, concurrently acquire resistance to another drug, B. On the other hand, collateral sensitivity is observed when bacteria, having developed resistance to drug C, show an increased susceptibility to another drug, A. When collateral sensitivity is established, it is anticipated that the simultaneous development of resistance to both drug A and drug C is challenging. This expectation suggests that combinations of these drugs could potentially hinder the development of bacterial resistance.

**Figure 2 antibiotics-13-00094-f002:**
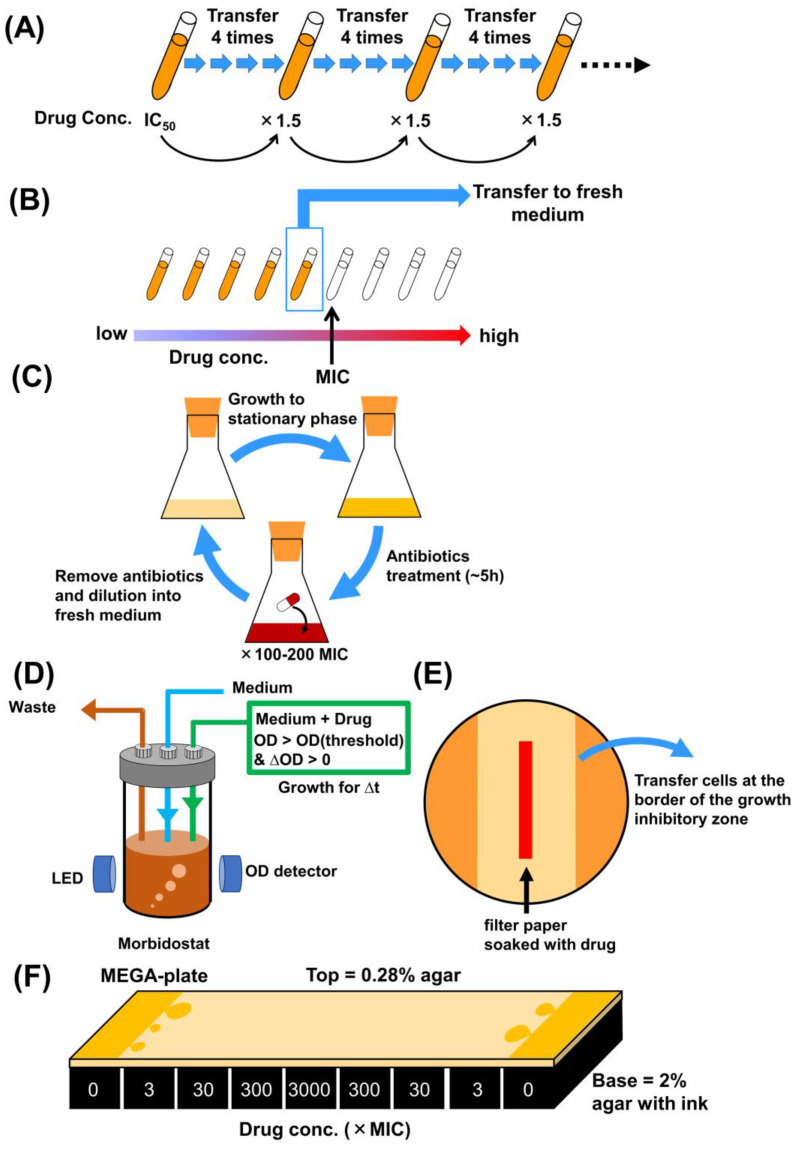
Methods of laboratory evolution. (**A**) Drug increment approach. During laboratory evolution, drug concentrations are regularly increased, e.g., 1.5-fold at every fourth transfer. (**B**) Drug gradient approach. Cells are grown in microwell plates containing drug gradients, and the culture that can grow at the highest drug concentration is selected and transferred to fresh drug gradient media. (**C**) Repeating batch culturing method between antibiotic treatment and the regrowth cycle for the acquisition of drug tolerance or persistence. (**D**) Morbidostat [39]. (**E**) Agar plate method under a drug gradient [40]. (**F**) MEGA-plate [23].

**Figure 3 antibiotics-13-00094-f003:**
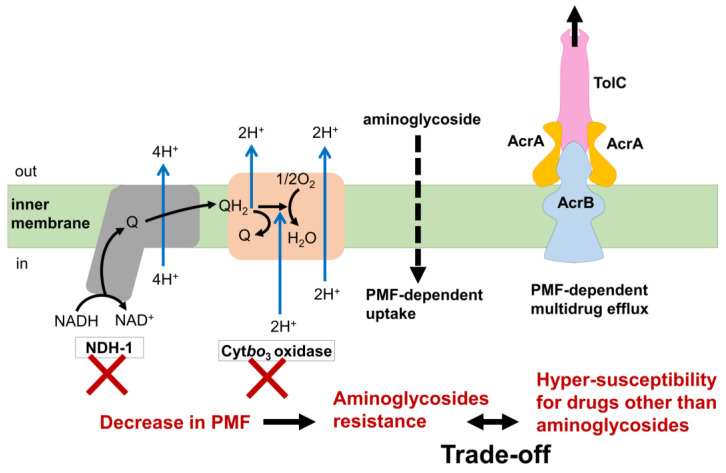
The trade-off mechanism of aminoglycoside resistance and drugs other than aminoglycosides. Q, quinone; QH_2_, quinol. *E. coli* possesses two isozymes with different abilities to generate PMF for NADH dehydrogenase and cytochrome oxidase in the respiratory chain. Aminoglycosides are known to be taken up into cells in a PMF-dependent manner. *E. coli* develops resistance to aminoglycosides through mutations that decrease PMF generation in either NDH-I or Cyt*bo*_3_ oxidase, both of which exhibit a high ability to generate PMF. In contrast, many antibiotics other than aminoglycosides develop resistance by enhancing the activity of PMF-dependent multidrug efflux systems like AcrAB/TolC. However, this resistance mechanism requires a high PMF. Therefore, a trade-off occurs between resistance to aminoglycosides and resistance to other drugs due to the contrasting PMF requirements in these two resistance pathways.

**Table 1 antibiotics-13-00094-t001:** Catalog of key genes conferring cross-resistance and collateral sensitivity in *E. coli*.

Mutation	Cross-Resistance	Collateral Sensitivity
*ompF*	Chloramphenicol, Rifampicin, Cefmetazole, Aztreonam, Carbenicillin, Norfloxacin, Phleomycin, DL-3-hydroxynorvaline, Mecillinam, Tetracycline, Furaltadone, Erythromycin, Puromycin	D-Cycloserine
*rssB*	Chloramphenicol, Rifampicin, Cefmetazole, Aztreonam, Acriflavine, Carbenicillin, Phleomycin, Tetracycline, Erythromycin, Puromycin	Protamine Sulfate, D-Cycloserine
*glpT*	Carbenicillin, Fosfomycin, Mitomycin C, Phleomycin, Puromycin	Protamine Sulfate, D-Cycloserine, Erythromycin
*cyaA*	Chloramphenicol, Rifampicin, Cefmetazole, Aztreonam, Acriflavine, Carbenicillin, Fosfomycin, Phleomycin, DL-3-hydroxynorvaline, Mecillinam, Tetracycline, Erythromycin, Puromycin	Vancomycin
*ycbZ*	Chloramphenicol, Aztreonam, Carbenicillin, Phleomycin, DL-3-hydroxynorvaline, Tetracycline, Erythromycin, Puromycin	D-Cycloserine
*cyoE*	Kanamycin, D-Cycloserine, Phleomycin, DL-3-hydroxynorvaline, Puromycin	Rifampicin, Erythromycin
*cyoA*	Kanamycin, Phleomycin	Vancomycin
*cyoB*	Kanamycin, D-Cycloserine, Phleomycin	Sulfisoxazole, Vancomycin
*nuoG*	Aztreonam, Carbenicillin, Phleomycin, DL-3-hydroxynorvaline, Tetracycline, Puromycin	Chloramphenicol
*mipA*	Acriflavine, Mitomycin C, Phleomycin, DL-3-hydroxynorvaline, Tetracycline, Puromycin	Vancomycin
*ptsP*	Aztreonam, Kanamycin, Phleomycin, DL-3-hydroxynorvaline, Puromycin	D-Cycloserine
*rfe*	Rifampicin, Carbenicillin, Mitomycin C, Phleomycin, Mecillinam	D-Cycloserine
*purR*	Carbenicillin, Phleomycin, DL-3-hydroxynorvaline, Puromycin	Sulfisoxazole, D-Cycloserine
*corA*	Chloramphenicol, Carbenicillin, Phleomycin, DL-3-hydroxynorvaline, Tetracycline, Puromycin	Sulfisoxazole
*oxyR*	DL-3-hydroxynorvaline	Norfloxacin
*apt*	Carbenicillin, Puromycin	D-Cycloserine
*sdaA*	Carbenicillin, Phleomycin, DL-3-hydroxynorvaline, Puromycin	D-Cycloserine
*nfsA*	Chloramphenicol, Aztreonam, Carbenicillin, Phleomycin, DL-3-hydroxynorvaline, Mecillinam, Nitrofurantoin, Furaltadone, Erythromycin, Puromycin	D-Cycloserine
*ilvL*	Chloramphenicol, Acriflavine, Carbenicillin, Puromycin	Sulfisoxazole
*gshA*	Chloramphenicol, Cefmetazole, Aztreonam, Acriflavine, Carbenicillin, Phleomycin, DL-3-hydroxynorvaline, Tetracycline, Erythromycin, Puromycin	Sulfisoxazole
*dacA*	Chloramphenicol, Acriflavine, Mitomycin C, Phleomycin	Cefmetazole, Erythromycin
*frlA*	Chloramphenicol, Acriflavine	Vancomycin
*fusA*	Chloramphenicol, Rifampicin, Kanamycin, Acriflavine, Carbenicillin, Sulfisoxazole	Protamine Sulfate, D-Cycloserine, Vancomycin
*glyT*	Chloramphenicol, Aztreonam, Acriflavine, Carbenicillin, Phleomycin, DL-3-hydroxynorvaline, Tetracycline, Erythromycin, Puromycin	Sulfisoxazole
*gyrA*	Chloramphenicol, Cefmetazole, Aztreonam, Carbenicillin, Norfloxacin, Tetracycline, Erythromycin, Puromycin	Acriflavine, Fosfomycin, D-Cycloserine
*hisS*	Chloramphenicol, Tetracycline	D-Cycloserine
*iscR*	Chloramphenicol, Carbenicillin, Mitomycin C, Puromycin	D-Cycloserin
*livM*	Chloramphenicol, Carbenicillin, Tetracycline\	Sulfisoxazole
*lon*	Chloramphenicol, Cefmetazole, Acriflavine, Carbenicillin, D-Cycloserine, Phleomycin, DL-3-hydroxynorvaline, Mecillinam, Tetracycline, Erythromycin, Puromycin	Mitomycin C
*rne*	Chloramphenicol, Rifampicin, Cefmetazole, Aztreonam, Acriflavine, Carbenicillin, Mitomycin C, Erythromycin	Sulfisoxazole, Vancomycin
*rob*	Chloramphenicol, Rifampicin, Cefmetazole, Aztreonam, Carbenicillin, Norfloxacin, Mitomycin C, Tetracycline, Erythromycin, Puromycin	D-Cycloserine
*rpoB*	Chloramphenicol, Carbenicillin, Mitomycin C, Amitriptyline, Tetracycline, Puromycin	D-Cycloserine
*serA*	Chloramphenicol, Aztreonam, Carbenicillin, Norfloxacin, Phleomycin, DL-3-hydroxynorvaline, Tetracycline, Puromycin	Fosfomycin, D-Cycloserine
*prlF*	Chloramphenicol, Aztreonam, Kanamycin, Carbenicillin, D-Cycloserine, Phleomycin, Mecillinam, Puromycin	Rifampicin
*yjcO*	Carbenicillin, Phleomycin, Tetracycline	Sulfisoxazole, D-Cycloserine
*acrR*	Chloramphenicol, Rifampicin, Cefmetazole, Aztreonam, Acriflavine, Carbenicillin, Mitomycin C, Tetracycline, Promethazine, Nitrofurantoin, Furaltadone, Erythromycin, Puromycin	D-Cycloserine

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
