# Peer review of "Laboratory Evolution of Antimicrobial Resistance in Bacteria to Develop Rational Treatment Strategies"

_antibiotics, 2024, doi:10.3390/antibiotics13010094_

Round 1

Reviewer 1 Report

Comments and Suggestions for Authors

Recommendation: Major Revision

Manuscript Number: antibiotics-2817893

Journal: Antibiotics

Title: Laboratory evolution of antimicrobial resistance in bacteria to develop rational treatment strategies

Overview and general recommendation:

This paper provides a comprehensive review of antibiotic resistance (AMR), discussing its global impact, mechanisms, and laboratory research methodologies. The information has an understandable and straightforward structure. It gives a clear picture of how resistance develops in laboratories and how to address it. The material is arranged logically and provides straightforward explanations of ideas like treatment plans and collateral sensitivity. Still, there are some recommendations to improve and simplify the information.

Comments:

1.      On page no. 2, Line no. 66-67. Authors should define collateral sensitivity and cross-resistance terms to help in making the content accessible to a broader audience.

2.      On page no 6, line no 193. There is a typo in "TThis study," and this should be corrected.

3.      On page no. 11, line no 458. The authors should provide a brief statement on tavaborole and its relevance.

4.      On pages 11-12,  section 4.2. Authors should address the practical applications of collateral sensitivity cycling in real-world scenarios with more examples.

5.      On page no 13, line no 559-574, section 4.5. The authors need to provide a few more points regarding the specific antibiotic mentioned and their metabolic dependencies.

6.      In conclusion section 5, the Authors should highlight the consequences of targeting specific mutations in clinical settings. How reasonable is this approach, and what challenges might arise?

7.      Authors should provide practical suggestions of the findings for the development of innovative therapeutic approaches more explicitly.

Author Response

We would like to thank you for the constructive suggestions to improve our manuscript. We have revised the manuscript following your suggestions. Our responses to specific comments are shown below.

Comment 1. On page no. 2, Line no. 66-67. Authors should define collateral sensitivity and cross-resistance terms to help in making the content accessible to a broader audience.

Response 1.

Thank you for the comment. As you suggested, we have corrected the part as follows:

P2 Lines 75-79

The development of AMR against a primary drug may lead to a dual outcome, inducing either heightened or reduced susceptibility to other drugs concurrently. This phenomenon is known as collateral sensitivity, wherein resistance to a particular drug increases susceptibility to another, or cross-resistance, wherein resistance to one drug amplifies resistance to another (Figure 1) [19].

Comment 2. On page no 6, line no 193. There is a typo in "TThis study," and this should be corrected.

Response 2.

Thank you for finding the typo. We have corrected the part as follows:

P7 Lines 236-238

This study revealed convergent molecular evolution across antibiotics, mutations conferring resistance enhancing sensitivity to other drugs, and the identification of chemogenomic profile similarity as a predictor of cross-resistance [21].

Comment 3. On page no. 11, line no 458. The authors should provide a brief statement on tavaborole and its relevance.

Response 3.

Thank you for the comment. As you suggested, we have provided a brief statement on tavaborole and its relevance as follows:

P15 Lines 536-551

Tavaborol, a novel synthetic small-molecule inhibitor of protein synthesis, received approval from the Food and Drug Administration in 2014 for the treatment of onychomycosis (nail fungus) [83–86]. Tavaborole functions by targeting the editing domain of leucyl-tRNA synthetase (LeuRS). Through covalent binding with tRNALeu, tavaborole prevents the dissociation of tRNALeu from LeuRS [84]. It has been demonstrated that tavabolore and its deliberatives are effective against multidrug-resistant bacterial pathgens, including E. coli, P. aeruginosa [86], S. aureus [87], Mycobacterium tuberculosis [88–91], and S. pneumoniae [92]. The resistant mechanism against tavaborole was elucidated through the acquisition of specific mutation in the LeuRS editing domain, as revealed by whole-genome sequencing of clinical isolates of E. coli [93] and P. aeruginosa [94]. Since the tavabole resistant E. coli strain carrying the LeuRS mutation showed hypersensitivity to norvaline (a chemical analog of leucine), the cost of tavabole resistance can be amplified in the presence of norvaline, which is misused for protein synthesis by the resistant cells [82]. Indeed, Melnikov et al. demonstrated that simultaneous tavaborole and norvaline treatment slowed tavaborole resistance evolution [82]. It is crucial to note that complete suppression was not achieved, indicating a potential risk of multidrug resistance [82].

Comment 4. On pages 11-12, section 4.2. Authors should address the practical applications of collateral sensitivity cycling in real-world scenarios with more examples.

Response 4.

Thank you for your valuable comment. As you suggested, we have added the descriptions addressing the practical applications of collateral sensitivity cycling in real-world scenarios in section 4.2 at lines 603-634. Also, we have added a new Table 2 to summarize the treatment strategies and their links with the literature knowledge as well as the pros and cons of these strategies. Please also see the new section 4.7 and Table 2.

P16 Lines 603-634

The impact of an antibiotic cycling strategy on the prevalence of antibiotic-resistant, Gram-negative bacteria was investigated through a cluster-randomized crossover study conducted across eight intensive care units (ICUs) in Europe [99]. Antibiotic-resistant, Gram-negative bacteria were operationally defined as Enterobacteriaceae with extend-ed-spectrum β-lactamase production or resistance to piperacillin–tazobactam, along with Acinetobacter spp and P. aeruginosa displaying resistance to piperacillin–tazobactam or carbapenems within the study parameters [99]. The study centered on three antibiotic groups—third-generation or fourth-generation cephalosporins, piperacillin–tazobactam, and carbapenems—and evaluated the prevalence of resistant bacteria during cycling (where a specific antibiotic class was preferentially employed for 6-week periods) and mixing (where the antibiotic class was altered for each consecutive patient) [99]. Despite observed variations in antibiotic utilization across ICUs, the comprehensive analysis of study antibiotics' overall volume and consumption patterns during cycling and mixing revealed no statistically significant differences [99]. The principal examination, accounting for patients present during monthly point-prevalence surveys, disclosed a mean prevalence of antibiotic-resistant, Gram-negative bacteria at 23% during cycling and 22% during mixing, with no discernible statistical distinction between the two strategies [99]. The study concluded that antibiotic cycling does not manifest a substantial reduction in the prevalence of antibiotic-resistant bacteria in ICUs [99]. It is noteworthy to mention that this study was conceived in 2010 [99], predating the development of collateral sensitivity cycling introduced by Imamovic et al. in 2013 [15]. Consequently, there exists a positive prospect regarding the clinical efficacy of this strategy; however, additional clinical evidence is imperative to adjudicate the reliability of this approach. To implement collateral sensitivity cycling in real-world scenarios, the collateral effects of clinically relevant drug combinations must exhibit robustness across genetically diverse backgrounds. Therefore, the robustness of collateral sensitivity in various clinical resistant strains has been intensively analyzed. For example, the study demonstrated the robustness of collateral sensitivity with ciprofloxacin towards gentamicin, fosfomycin, ertapenem, and colistin in E. coli [100], and towards gentamicin, fosfomycin, colistin, aztreonam, and tobramycin in P. aeruginosa [25][101]. The efficacy of aminoglycoside and β-lactam in eradicating quinolone-resistant P. aeruginosa from patients suffering from cystic fibrosis was also demonstrated [25].

Comment 5. On page no 13, line no 559-574, section 4.5. The authors need to provide a few more points regarding the specific antibiotic mentioned and their metabolic dependencies.

Response 5.

Thank you for your comment. As you suggested we have provided a few more points regarding the specific antibiotic mentioned and their metabolic dependencies as follows:

P18 Lines 692-702

The metabolic dependency of E. coli was assessed by examining the bactericidal effectiveness of various antibiotics across a range of nutrient availabilities [103]. This investigation revealed distinct categories of antibiotics regarding their reliance on metabolism, with ampicillin and ciprofloxacin identified as strongly dependent, while gentamicin, halicin, and mitomycin C were categorized as weakly dependent [103]. Notably, ampicillin and ciprofloxacin exhibited diminished bactericidal efficacy when faced with nutri-ent-depleted cells, in contrast to mitomycin C, which maintained their effectiveness under these conditions [103]. The effectiveness of strongly dependent drugs exhibited a positive correlation with increased intracellular ATP concentration (and therefore metabolic state). Conversely, the effectiveness of weakly dependent drugs appeared largely unaffected by variations in intracellular ATP concentration [103].

P18 Lines 705-708

Tolerance evolved more readily against antibiotics strongly dependent on bacterial metabolism such as ampicillin and ciprofloxacin compared to those minimally affected by metabolic state such as gentamicin, halicin, and mitomycin C [107].

Comment 6. In conclusion section 5, the Authors should highlight the consequences of targeting specific mutations in clinical settings. How reasonable is this approach, and what challenges might arise?

Response 6.

Thank you for your valuable comments. To answer your comments, we have added a new section at 4.7 entitled “Clinical evidence supporting the efficacy of antibiotic combination therapy involving aminoglycosides is substantial”, instead of adding the explanations in the conclusion section 5. In addition, we have added a new section 3.2 entitled “Identified key genes conferring cross-resistance and collateral sensitivity in E. coli” to provide more detailed explanations of the identified key genes and their functional significance in the context of cross-resistance and collateral sensitivity. The key genes associated with these phenomena in E. coli have been cataloged in Table 1. Please see the new sections 4.7, 3.2, and Table 1.

Comment 7. Authors should provide practical suggestions of the findings for the development of innovative therapeutic approaches more explicitly.

Response 7.

Thank you for your valuable comment. We have added practical suggestions in the conclusion section 5 as follows:

P21 Lines 844-856

Recognizing that AMPs and phages serve as viable alternatives to antibiotics, effective combinations should extend beyond antibiotics alone. The design of combinations should not be confined solely to antibiotics. Presently, the development of an ideal drug capable of completely suppressing resistance acquisition remains elusive. Similar to antibiotics, reports of resistance acquisition to AMPs and phages exist [120,122]. The distinct pros and cons associated with antibiotics, AMPs, and phages [123,124] offer endless possibilities for effective combinations. It is important to highlight that the use of aminoglycosides in combinations may pose a risk of notable side effects, potentially surpassing the clinical benefits [109,114]. Consequently, there is a strong need to identify drug pairs that exhibit evolutionary trade-offs, distinct from the reliance on aminoglycosides. To identify ideal combinations based on their evolutionary constraints, further high-throughput laboratory evolution will be necessary to establish a robust starting point.

Reviewer 2 Report

Comments and Suggestions for Authors

In the manuscript “Laboratory evolution of antimicrobial resistance in bacteria to develop rational treatment strategies”, the authors described the laboratory methods in studying bacterial evolution in antibiotic resistance and gave a comprehensive review of knowledge learned from literatures and their potential use to design treatment strategies to combat antibiotic resistance. This reviewer has a few minor comments.

1.    The paper should explain the terms in context in addition to their fully names when they firstly appeared in the manuscript, such as MIC and DFE.

2.    In addition to the laboratory evolution strategies, many techniques and concepts were used in the literatures covered by the manuscript and some of them need to be described in detail for understanding these studies. For example, the study mentioned in LINE 220-238, it’s unclear for how the study was done and what are the “low-dimensional phenotypic states and “intracellular states” defined by the corresponding study. In section 3.7, before going into deeper discussions, the paper should first explain the fitness landscape in the context of the study. Other terms such as transcriptomes and metabolome should also be explained. The paper could either add another section to introduce these techniques/concepts or explain them in the places where they were firstly mentioned.

3.    The paper could benefit from adding a table or a graph to summarize the treatment strategies and their links with the literature knowledge as well as the pros and cons of these strategies when applicable.

4.    The study could briefly summarize the current other options/attempts for combating antibiotics resistance besides using treatment strategies with antibiotics.

Author Response

We would like to thank you for the constructive suggestions to improve our manuscript. We have revised the manuscript following your suggestions. Our responses to specific comments are shown below.

Comment 1. The paper should explain the terms in context in addition to their fully names when they firstly appeared in the manuscript, such as MIC and DFE.

Response 1.

Thank you for your comment. As you suggested, we have corrected throughout the manuscript.

For example,

P1-2 Lines 39-42

Cells resistant to antibiotics exhibit an elevation in the minimal inhibitory concentrations (MICs), which denotes the lowest antibiotic concentration needed to hinder bacterial replication. In contrast, antibiotic persister cells and tolerant cells demonstrate enhanced survival under antibiotic exposure without a concurrent rise in the MIC [5].

P10-11 Lines 346-349

DFE of mutations refers to the range and characteristics of fitness changes that result from genetic mutations within a population. The DFE is thought to be a crucial factor influencing the evolutionary dynamics of drug resistance in bacterial populations [54].

Comment 2. In addition to the laboratory evolution strategies, many techniques and concepts were used in the literatures covered by the manuscript and some of them need to be described in detail for understanding these studies. For example, the study mentioned in LINE 220-238, it’s unclear for how the study was done and what are the “low-dimensional phenotypic states and “intracellular states” defined by the corresponding study. In section 3.7, before going into deeper discussions, the paper should first explain the fitness landscape in the context of the study. Other terms such as transcriptomes and metabolome should also be explained. The paper could either add another section to introduce these techniques/concepts or explain them in the places where they were firstly mentioned.

Response 2.

Thank you for your valuable comments. As you suggested, we have corrected the manuscript. The major correction is that we have added new paragraphs to introduce such techniques/concepts in the introduction section 1 as follows.

P2-3 Lines 87-118

The "landscape" metaphor in evolutionary biology refers to a visual representation of how different traits or characteristics of organisms (in this case, microbial resistance to antibiotics) change over time [28,29]. It is dynamic because the conditions that microbes face, such as exposure to antibiotics or other environmental factors, are constantly shifting. Just as a landscape can have peaks and valleys, the dynamic landscape of resistance illustrates how microbial traits might rise or fall in response to various selection pressures.

The concept of the "dimensionality of phenotypic states" in the context of AMR evolution discusses how traits or characteristics of microbes (like their ability to resist drugs) can be expressed in different ways [30,31]. This entails exploring how these traits can change and adapt over time in response to factors like drug exposure. There are numerous components or features in microbial cells, often measured as omics data such as transcriptome (the set of all RNA transcripts), proteome (the entire set of proteins), metabolome (the complete set of intracellular small-molecule chemicals), and genomic mutations (the loci and the type of mutations) [22,27,32–34]. These cellular components contribute to a large number of degrees of freedom, forming high-dimensional data. Despite this complexity, changes related to adaptation and evolution were observed as effectively limited to a lower-dimensional subspace [22,27]. This implies that, even though there are many variables, only a few of them are crucial for expressing traits that impact the microbe's ability to evolve and adapt, especially in the context of AMR. Therefore, certain traits related to AMR evolution can be represented or changed in simpler, more predictable ways during evolution by focusing on a few key aspects rather than the entire complexity of the microbe's characteristics. Simplifying this high-dimensional data into a lower-dimensional representation helps bridge the gap between the intricate details of cellular components and the essential biological information, such as growth rate, adaptability, and survival. The dimension reduction implies that phenotypic changes are constrained to a low-er-dimensional subspace, meaning that certain key traits change more significantly than others [30,31,35]. Once these constraints are acquired through evolution, subsequent adaptations to new environments tend to follow the same restricted paths [35]. These concepts are crucial for understanding how microbes adapt and evolve in response to challenges like antibiotic exposure. The focus on lower-dimensional representations helps identify key factors that play a major role in adaptation, making it easier to predict and potentially control the evolutionary dynamics of AMR.

Comment 3. The paper could benefit from adding a table or a graph to summarize the treatment strategies and their links with the literature knowledge as well as the pros and cons of these strategies when applicable.

Response 3.

Thank you for your valuable comment. As you suggested, we have added a new Table 2 entitled “Three evolutionary-based strategies to combat AMR” to summarize the treatment strategies and their links with the literature knowledge as well as the pros and cons. Please see the new Table 2.

Comment 4. The study could briefly summarize the current other options/attempts for combating antibiotics resistance besides using treatment strategies with antibiotics.

Response 4.

Thank you for your valuable comment. As you suggested, we have introduced the use of antimicrobial peptides (AMPs) as the current other options/attempts besides using treatment strategies with antibiotics. We have added a new section 4.8 entitled “Application of antimicrobial peptides (AMPs) for combating AMR”.

P19-20 Lines 778-818

4.8 Application of antimicrobial peptides (AMPs) for combating AMR

The emergence of drug resistance to conventional antibiotics necessitates the exploration of alternative agents, such as AMPs. AMPs demonstrate broad-spectrum antimicrobial activity characterized by high specificity and low toxicity, derived from both synthetic and natural sources [118]. They play a pivotal role in the host's innate immunity against various microorganisms, including bacteria, fungi, parasites, and viruses [118]. AMPs are considered promising alternatives due to their capacity to act on multiple targets, com-bating drug-resistant bacteria [118]. Therefore, current clinical applications of AMPs primarily focus on bacterial infections. Cationic AMPs, effective against bacterial pathogens, target unique anionic components such as LPS in Gram-negative bacteria and lipoteichoic acid in Gram-positive bacteria [118]. The structural diversity of AMPs and their ability to target specific lipids or intracellular proteins contribute to their selectivity against bacterial species or strains. AMPs induce membrane permeation, leading to intracellular content leakage, or penetrate membranes for intracellular effects [118]. In the context of bio-films—microbial aggregates forming on tissues or medical implants—AMPs offer diverse mechanisms to target various biofilm properties [119].

A systematic examination of resistance evolution to 14 diverse cationic AMPs and 12 antibiotics in E. coli was conducted [120]. Laboratory evolution of E. coli against these AMPs and antibiotics demonstrated that certain AMPs, such as tachyplesin II and R8, completely suppress resistance acquisition during approximately 120 generations of laboratory evolution, sufficient time for the acquisition of resistances to the 12 antibiotics [120]. Tachyplesins I and II, antimicrobial peptides showing potent activity against various pathogens, exhibited no observed resistance acquisition in a mutator E. coli strain and pathogens including S. enterica subsp. serovar Typhimurium, K. pneumoniae subsp. pneumoniae, and A. baumannii, indicating a very low probability of resistance acquisition against tachyplesin II [120]. Increased hydropathicity and fewer polar and positively charged amino acids were associated with reduced resistance during laboratory evolution [120]. In contrast to AMR evolution, no cross-resistance to AMPs and limited horizontal gene transfer for AMP resistance were observed [120]. The possibility of AMP resistance through horizontal gene transfer was assessed by heterologous expression of meta-genomic DNA fragments from soil samples (1.8 million clones) in E. coli [120].

The effectiveness of combinations of AMPs on AMP evolution in S. aureus was also explored [121]. It was demonstrated that treatment with certain AMP combinations delays resistance evolution compared to individual AMPs [121]. The lowest resistance was observed with a random AMP library containing over a million peptides [121]. The study highlights the correlation between resistance evolution rate, individual AMP resistance cost, and cross-resistance. AMP-resistant strains often remain sensitive to other AMPs, reducing concerns about broad-range resistance [121]. Combining AMPs, especially those with high resistance costs, proves effective in hindering resistance evolution, emphasizing the potential of AMP cocktails for sustainable treatment against antibiotic-resistant pathogens.

Reviewer 3 Report

Comments and Suggestions for Authors

Laboratory evolution studies, particularly focusing on Escherichia coli, have provided valuable insights into the mechanisms of antimicrobial resistance (AMR). These investigations demonstrate that repetitive exposure to antibiotics leads to bacterial populations adapting and developing tolerance and resistance to drugs. The studies reveal convergent evolution across different antibiotics, the pleiotropic effects of resistance mutations, and the influence of loss-of-function mutations. Genotype-phenotype explorations have identified key genes predicting drug resistance, and quantitative analyses of multidrug combinations have uncovered collateral sensitivity. Some drug combinations with collateral sensitivity show the potential to suppress the acquisition of resistance. The review outlines the methodologies employed in laboratory evolution studies of AMR and highlights recent discoveries concerning AMR mechanisms. It also discusses the application of laboratory evolution in developing rational treatment strategies.

Strengths:

1.       The review provides a comprehensive overview of laboratory evolution studies, emphasizing their contribution to understanding antimicrobial resistance mechanisms.

2.       The inclusion of convergent evolution, pleiotropic effects, and the role of loss-of-function mutations adds depth to the discussion.

Weaknesses:

    1. While the review mentions key genes predicting drug resistance, more details about these genes and their specific roles in resistance mechanisms could enhance the understanding of the genetic aspects of AMR. Specifically, the abstract needs to be modified.
    2. The review could benefit from specific examples or case studies to illustrate the practical applications of laboratory evolution in formulating treatment strategies.

Suggestions

    1. Elaborating on the practical applications of laboratory evolution in the development of rational treatment strategies would enhance the relevance and impact of the review.
    2. Providing more detailed explanations of the identified key genes and their functional significance in the context of drug resistance would strengthen the genetic aspect of the discussion.

Overall, the review effectively synthesizes information on laboratory evolution studies and their insights into antimicrobial resistance mechanisms. Adding more details about key genes and practical applications could further enhance the review's depth and relevance.

Comments on the Quality of English Language

none

Author Response

We would like to thank you for the constructive suggestions to improve our manuscript. We have revised the manuscript following your suggestions. Our responses to specific comments are shown below.

Comment 1. While the review mentions key genes predicting drug resistance, more details about these genes and their specific roles in resistance mechanisms could enhance the understanding of the genetic aspects of AMR. Specifically, the abstract needs to be modified.

Response 1.

Thank you for your comment. As you suggested, we have corrected the abstract as well as the correction of the manuscript according to your further comments. Please see the next responses 2-3.

Abstract (P1 Lines 10-26)

Laboratory evolution studies, particularly with Escherichia coli, have yielded invaluable insights into the mechanisms of antimicrobial resistance (AMR). Recent investigations have illuminated that with repetitive antibiotic exposures, bacterial populations will adapt and eventually become tolerant and resistant to the drugs. Through intensive analyses, these inquiries have unveiled instances of convergent evolution across diverse antibiotics, elucidated the pleiotropic effects of resistance mutations, and the role played by loss-of-function mutations in the evolutionary landscape. Moreover, a quantitative analysis of multidrug combinations has shed light on collateral sensitivity, revealing specific drug combinations capable of suppressing the acquisition of resistance. This review article introduces the methodologies employed in the laboratory evolution of AMR in bacteria and presents recent discoveries concerning AMR mechanisms derived from laboratory evolution. Additionally, the review outlines the application of laboratory evolution in endeavors to formulate rational treatment strategies.

Comment 2. The review could benefit from specific examples or case studies to illustrate the practical applications of laboratory evolution in formulating treatment strategies.

Suggestions

Elaborating on the practical applications of laboratory evolution in the development of rational treatment strategies would enhance the relevance and impact of the review.

Response 2.

Thank you for your valuable comment. To answer your comment, we have added a new section at 4.7 entitled “Clinical evidence supporting the efficacy of antibiotic combination therapy involving aminoglycosides is substantial”. Also, we have added a new Table 2 entitled “Three evolutionary-based strategies to combat AMR” to summarize the treatment strategies and their links with the literature knowledge as well as clinical evidence and the pros and cons. Please see the new Table 2.

P19 Lines 740-776

4.7 Clinical evidence supporting the efficacy of antibiotic combination therapy involving aminoglycosides is substantial.

The utilization of such combinations has become widespread, particularly in addressing severe hospital-acquired infections caused by multidrug-resistant species, owing to the apparent effectiveness of the evolutionary trade-off relationships between aminoglycoside resistance and other drugs [109–111]. A retrospective study focusing on bacteremia predominantly induced by Enterobacter cloacae in cancer patients was conducted by Bodey et al., in 1991 [112]. The study assessed response rates, considering eradication of all signs and symptoms of Enterobacter infection as the endpoint. Results indicated response rates of 59% and 74% with the single use of aminoglycoside or penicillin, respectively, while simultaneous administration of aminoglycoside and penicillin demonstrated a higher response rate of 78% [112]. Examining the clinical efficacy and safety of combination therapy involving aminoglycoside antibiotic gentamicin and macrolide antibiotic azithromycin for treating urogenital gonorrhea caused by Neisseria gonorrhoeae infection, a study conducted by Kirkcaldy et al., in 2014 reported promising outcomes [113]. The randomized, multisite, open-label, noncomparative trial revealed a 100% microbiological cure rate among 202 evaluable participants receiving gentamicin/azithromycin [113]. Nevertheless, there exist discrepancies regarding the use of such two-combination therapy [109]. For instance, a randomized trial comparing β-lactam monotherapy with β-lactam-aminoglycoside combination therapy for sepsis in immunocompetent patients, inclusive of various pathogens such as S. aureus, Enterobacteriaceae, and P. aeruginosa, concluded that the addition of an aminoglycoside to β-lactams is discouraged due to unaltered fatality rates [114]. A subsequent study corroborated this conclusion, however, this study additionally concluded a survival benefit for β-lactam-aminoglycoside combination therapy when prescribed for children with multidrug-resistant Gram-negative bacteria [110].

Despite lingering uncertainty, the use of aminoglycosides in combination therapy has been recommended, emphasizing risk stratification [111][115]. It is crucial to note that prior combination therapies were designed based on the synergistic effects of antibiotic combinations [109]. Recent laboratory evolution experiments, however, challenge this perspective by demonstrating that the evolvability of AMR remains independent of whether the combinations exhibit synergistic effects [21,59,116,117]. Rather than relying on observed drug interactions, it revealed a discernible pattern linking genetic trajectories to resistance evolution [116]. Therefore, future endeavors must delve into elucidating the mechanisms of evolutionary constraints and identifying specific drug pairs to inform the design of rational treatment strategies for combination therapy.

Comment 3. Providing more detailed explanations of the identified key genes and their functional significance in the context of drug resistance would strengthen the genetic aspect of the discussion.

Overall, the review effectively synthesizes information on laboratory evolution studies and their insights into antimicrobial resistance mechanisms. Adding more details about key genes and practical applications could further enhance the review's depth and relevance.

Response 3.

Thank you for your valuable comments. To answer your comments, we have added a new section 3.2 entitled “Identified key genes conferring cross-resistance and collateral sensitivity in E. coli” to provide more detailed explanations of the identified key genes and their functional significance in the context of cross-resistance and collateral sensitivity. The key genes associated with these phenomena in E. coli have been cataloged in Table 1. Please see the new Table 1.

P8 Lines 304-331

3.2  Identified key genes conferring cross-resistance and collateral sensitivity in E. coli

Our comprehensive high-throughput laboratory evolution of E. coli systematically investigated the underlying mechanisms of cross-resistance and collateral sensitivity [27]. The key genes associated with these phenomena in E. coli have been cataloged in Table 1, drawing upon insights from our previous study. Notably, the study emphasized the pivotal role of mutations in genes governing transporters and porins in mediating antibiotic resistance in E. coli [27]. Perturbations in uptake and efflux activities emerged as principal mechanisms governing cross-genetic resistance and heteroresistance [27]. Specifically, the study illuminated the significance of the overexpression of efflux pumps AcrAB/TolC and EmrAB/TolC, coupled with the inactivation of their repressors, in conferring resistance to a spectrum of antibiotics [27]. Furthermore, the investigation pinpointed the involvement of YcbZ, a putative protease implicated in translation and ribosome biogenesis, in mediating cross-resistance against multiple antibiotics [27]. Mutations in PrlF, associated with the PrlF-YhaV toxin-antitoxin system, were found to be linked to resistance against specific antibiotics, such as aztreonam and carbenicillin [27]. The observed cross-resistance was partly ascribed to the diminished expression of OmpF, underscoring the intricate interplay of genes in the stress response [27]. Additionally, the study uncovered collateral sensitivities tied to prlF-mediated resistance, revealing trade-offs between acquired resistances and susceptibility to specific drugs, including rifampicin [27]. It was postulated that these collateral sensitivities might be linked to the global mRNA destabilization effect induced by increased RNase activity of PrlF [27]. Moreover, the derepression of an alternative sigma factor, RpoS, resulting from a mutation in the regular RssB, conferred both cross-resistance and collateral sensitivity to various drugs [27]. The proposed mechanism for such collateral sensitivity involved the competition between RpoS and the housekeeping sigma factor RpoD, leading to decreased carbon source availabilities and diminished competitiveness for low concentrations of nutrients [27]. The subsequent section, 3.5, elaborates on the trade-off mechanism of aminoglycoside resistance and drugs unrelated to aminoglycosides.

Round 2

Reviewer 1 Report

Comments and Suggestions for Authors

The manuscript is recommended for acceptance.